

# Orb-weaving spiders are fewer but larger and catch more prey in lit bridge panels from a natural artificial light experiment

Dylan G.E. Gomes

Department of Biological Sciences, Boise State University, Boise, ID, United States of America

## ABSTRACT

Artificial light at night is rapidly changing the sensory world. While evidence is accumulating for how insects are affected, it is not clear how this impacts higher trophic levels that feed on insect communities. Spiders are important insect predators that have recently been shown to have increased abundance in urban areas, but have shown mixed responses to artificial light. On a single bridge with alternating artificially lit and unlit sections, I measured changes in the orb-weaving spider *Larinioides sclopetarius* (Araneidae) web abundance, web-building behavior, prey-capture, and body condition. In artificially lit conditions, spiders caught more prey with smaller webs, and had higher body conditions. However, there were fewer spiders with active webs in those lit areas. This suggests that either spiders were not taking advantage of an ecological insect trap, perhaps due to an increased risk of becoming prey themselves, or were satiated, and thus not as active within these habitats. The results from this natural experiment may have important consequences for both insects and spiders in urban areas under artificial lighting conditions.

## INTRODUCTION

Artificial light at night (ALAN) is an increasingly widespread pollutant that has considerable physiological and behavioral effects on wildlife, making it an urgent, worldwide conservation concern (*Longcore & Rich, 2004*; *Hölker et al., 2010*; *Gaston et al., 2013*; *Falchi et al., 2016*). Exposure to ALAN can alter circadian rhythms (*Chittka, Stelzer & Stanewsky, 2013*), inhibit reproduction (*Bebas, Cymborowski & Giebultowicz, 2001*; *Van Geffen et al., 2015*), accelerate development (*Van Geffen et al., 2015*; *Kehoe et al., 2018*), and lead to increased mortality (*Eisenbeis, 2006*).

It is well known that many groups of insects are attracted to artificial lights, often until exhaustion and death (*Horváth et al., 2009*), which is associated with large reductions in insect abundances in adjacent dark areas (*Eisenbeis, 2006*). While changes in animal communities due to ALAN have been documented (*Davies, Bennie & Gaston, 2012*; *Meyer & Sullivan, 2013*; *Manfrin et al., 2017*), our understanding of the indirect effects of ALAN on predators is somewhat limited.

Corresponding author
Dylan G.E. Gomes,
dylangomes@u.boisestate.edu

Many insect prey are attracted to light (reviewed in *Owens & Lewis, 2018*), and some potential predators may be exploiting this by foraging near lights (*Jung & Kalko, 2010*; *González-Bernal et al., 2016*) or by using light to forage into the night (*Russ, Rüger & Klenke, 2015*). Intermediate predators, such as spiders, have received little attention in the light pollution literature. Yet, as both predators of insects and prey themselves, spiders have the potential to dramatically alter community structure via top-down (*Riechert & Lockley, 1984*; *Nyffeler & Birkhofer, 2017*) and bottom-up forces (*Guinan & Sealy, 1987*; *Baxter, Fausch & Carl Saunders, 2005*; *Pagani-Núñez et al., 2011*).

The effects of artificial light on orb-weaving spiders of the Family Araneidae, has received some attention, but with conflicting results. There is both evidence of benefits (e.g., higher prey capture rates and increased abundance; *Heiling, 1999*) and costs (e.g., reduced body size, smaller clutch sizes, increased daily mortality; *Willmott et al., 2018*) of light to orb-weaving spiders. Yet, in these two experiments, different types of lights were used, and one study was primarily a field-based while the other was a laboratory study. Similarly, Araneid spiders demonstrate both positive (*Heiling, 1999*; *Willmott et al., 2019*) and aversive (*Yamashita & Tuji, 1987*; *Nakamura & Yamashita, 1997*) responses to light in choice tests, however these differences are likely due to species-specific responses and variation in light spectra.

Similarly, the effects of artificial lights on web-building behavior are unclear. Araneid spiders in the field make smaller webs in naturally-moonlit areas (*Adams, 2000*). This is likely because moonlight attracts prey, which satiates spiders, and satiated spiders build smaller webs (*Adams, 2000*; *Herberstein, Craig & Elgar, 2000*). Furthermore, previous prey capture success can alter other web building behaviors such as altering thread tension (*Watanabe, 2000*) or vertical asymmetry (*Heiling & Herberstein, 1999a*). In at least one laboratory study, artificial sources of light have made no difference in the size of webs (*Willmott et al., 2019*), yet artificial lighting can affect the frequency or propensity to build webs in the lab (*Zschokke & Herberstein, 2005*). However, web building is known to differ between laboratory and field settings in general (*Sensenig et al., 2010*; *Hesselberg, 2013*). Thus, it is still unclear how web building behavior changes in artificially lit systems in the wild.

Here I describe a natural experiment, a case study on a bridge in France, that investigates whether orb-weaving spiders, *Larinioides sclopetarius* (Family: Araneidae), avoid light, change web-building behavior under lit conditions, and whether potential changes in prey capture success alters body condition. If spiders are acting as optimal foragers, they should take advantage of artificial lights as ecological insect traps by being more active in lit areas. Spiders in lit areas should catch the more available prey (light-attracted insects) and, thus, have higher body condition.

## MATERIALS & METHODS

### Field location and experimental design

I collected data between August 13–23 of 2018 on the Saint-Symphorien suspension bridge over the Loire River in Tours, France (47.399158°N, 0.692519°E) between the hours of 2200

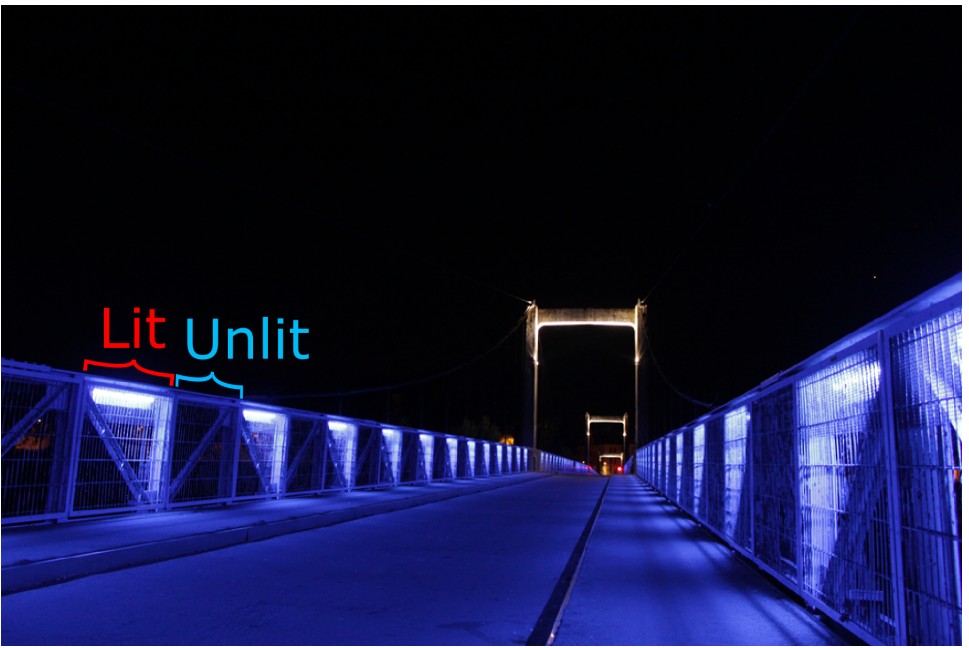

**Figure 1** **Saint-Symphorien suspension bridge in Tours, France where the natural experiment was conducted.** The image shows the block-on, block-off unintended design of the bridge, which allowed me to test the effects of artificial light on a continuous population of orb-weaving spiders that build webs within the handrail panels of the bridge.

and 0400. This 350 m pedestrian bridge consists of 324 individual metal side railing panels measuring 1.45 m × 1.15 m within which orb-weaving spiders (*Larinioides sclopetarius;* Family Araneidae) constructed webs. Bridge panels had an alternating lighting pattern; every other panel was lit from above by a blue fluorescent tube light (designated "lit"), which was absent from the other panels (designated "unlit") (Fig. 1). These two conditions acted as a natural block-on block-off design within which I measured spider abundance, web building behavior, and body condition. Bridge panel lights came on just before sunset (around 2100 during the days of the study), and remained on for the duration of the night. Data collection did not start until at least one hour after sunset.

## Spider web abundance

I measured spider web abundance within 88 of these panels (*n* = 44 lit, 44 unlit), which were treated statistically as independent groups within this bridge. Since these groups are not truly independent, these data should not be used to make generalizations about the population of *L. sclopetarius* at large, but are rather limited in scope to this bridge in particular. Potential issues of spatial autocorrelation were controlled for with the experimental design of the treatments, since lit and unlit panels were balanced across the bridge in an alternating pattern. That is, just as many panels in the middle (or sides) of the bridge were lit and unlit. I characterized web abundance via presence of 'active' webs—that is, if a female orb-weaving spider was positioned on the central hub or touching a radius of an orb web, it was considered an active predator on the landscape, and was included
in web counts. All abandoned orb-webs or Araneid spiders that were not associated with an orb-web were not counted. I visually identified spiders in the field, and a subset ($n = 38$) were collected and identified with a dichotomous key to species to ensure field identifications were correct (*Nentwig et al., 2016*). All spiders were identified as *Larinioides sclopetarius* (Family Araneidae).

## Web structure

I measured 86 webs (lit $n = 43$; unlit $n = 43$) as follows. I measured a maximum of one web per bridge panel, to aid in independent sampling. To avoid bias, I selected undamaged webs (no sections or radii of the orb web were destroyed or appeared to have been rebuilt) of mature adults in the order of observer discovery. That is, only the first web that was discovered, which met the above criteria, was selected for measurement.

I measured the vertical diameter ($D_v$), the horizontal diameter ($D_h$), the diameter of the free zone (the area lacking sticky capture silk; $D_{fz}$), and the radius of the top half of the web from the hub upward ($R_u$) of each web with a measuring tape to the nearest 0.5 cm (see Fig. 2). The lower radius ($R_l$) was calculated by subtracting $R_u$ from $D_v$. All measurements were taken from the center of the web (see *Tew & Hesselberg, 2017*). The overall web capture area was then calculated from the Ellipse-Hub equation (*Herberstein & Tso, 2000*):

$$Web\ area = \pi \left( \frac{D_v}{2} \right) \left( \frac{D_h}{2} \right) - \pi \left( \frac{D_{fz}}{2} \right)^2$$

Vertical web asymmetry was calculated with the following equation (*Tew & Hesselberg, 2017*):

$$Vertical\ web\ asymmetry = \frac{(R_u - R_l)}{(R_u + R_l)}.$$

## Prey capture and body condition

I counted the total number of prey items in a randomly-selected subset of webs (lit $n = 26$, unlit $n = 27$) to quantify capture success. To control for the effects of prey accumulation throughout the night, prey were counted on webs in lit and unlit conditions in an alternating pattern. That is, prey capture success measurements (between the two light conditions) were evenly spread across the night. All visually-identified prey captured by spiders were flies in the Family Chironomidae. Yet, since each prey was not identified with a microscope, it is possible that insects of other Families and Orders were captured as well. For this reason, I will retain the general use of the word 'prey' rather than being more specific.

Each captured spider was weighed to the nearest 0.001 grams with a digital scale, and I measured front-right femur lengths (Fe) with calipers to the nearest 0.05 mm. Body condition was calculated with the residual index, because it is thought to be the most reliable index for measuring body condition in spiders (*Jakob, Marshall & Uetz, 1996*; *Schulte-Hostedde et al., 2005*). I first regressed spider weight against femur length (Wt~Fe). Then I used the '*resid*' {*stats*} function in the programming language, R (version 3.5.1), to

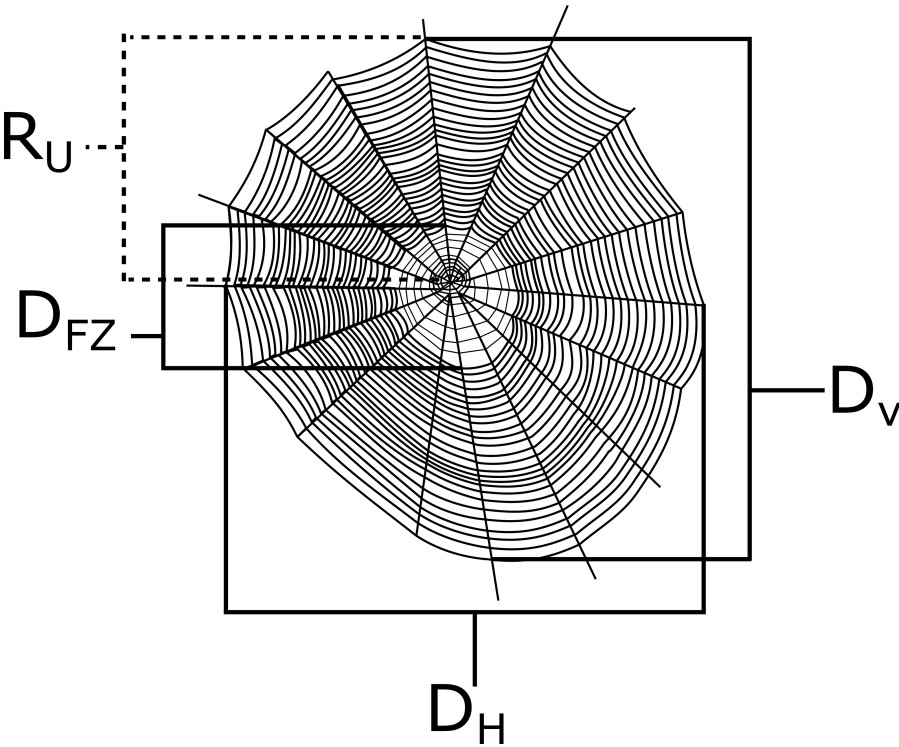

**Figure 2** **A depiction of a generic orb web with example measurements that were taken in this study.**
The horizontal diameter ($D_h$), vertical diameter ($D_v$), diameter of the free zone ($D_{fz}$), and the radius of the top half of the web ($R_u$) are all shown here. Webs were measured along the center of the webs.

calculate residual distances from the regression line. Positive residual values are indicators of 'good' body condition, while negative residual values are considered 'poor' body condition. Thus, I used these residuals as body condition metrics in further analyses (*Jakob, Marshall & Uetz, 1996*; *Schulte-Hostedde et al., 2005*).

## Statistics

All data were analyzed with either a linear model (LM) or a generalized linear model (GLM) in version 3.5.1 of R (*R Core Team, 2017*). Spider web abundance and prey counts were analyzed with GLMs, with a negative binomial distribution (log link; package 'glmmTMB'; *Magnusson et al., 2017*) because spider webs, and the prey within, are both count data, were over-dispersed with right skew, and were much better fitting models (residual plots) than LMs (*Bates, 2010*). Spider web area had a similar-shaped distribution, yet with continuous data. Thus these data were analyzed with a Gamma distribution and log link in a GLM (function 'glm' in R). Vertical web asymmetry, and body condition were analyzed using linear models (function 'lm' in R). All model residuals were checked with the function 'qqplot' or with the package 'DHARMa' (*Hartig, 2019*). Vertical web asymmetry can be affected by web size (*Tew & Hesselberg, 2018*), thus I used web area (size) along with light condition as predictors in the linear model to explain vertical web asymmetry. Similarly,

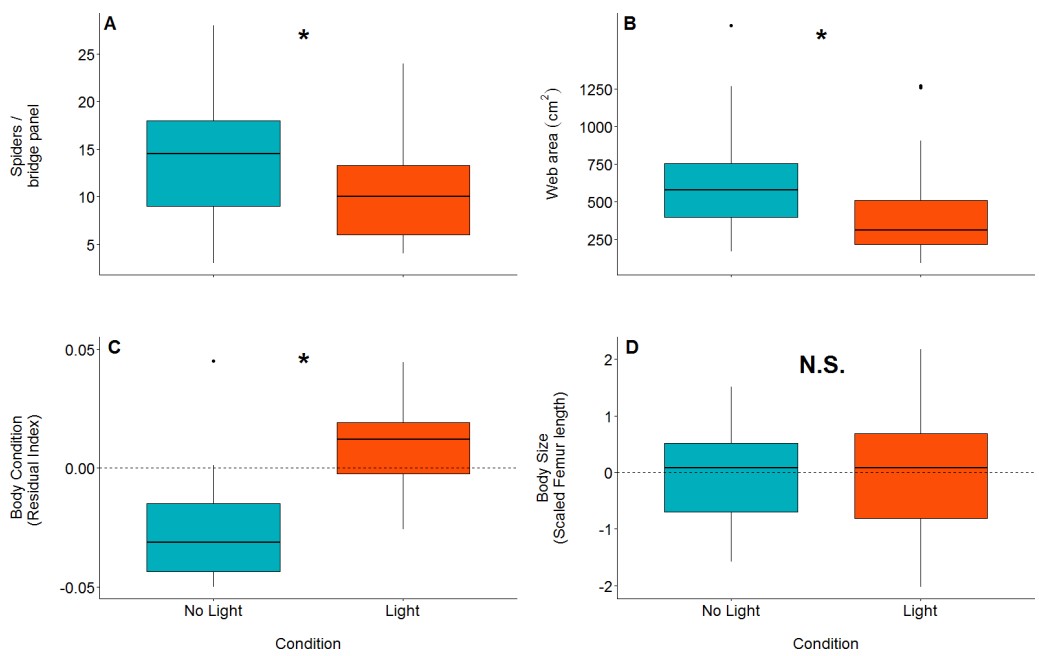

**Figure 3** **Boxplots of differences between measured variables in lit (red) and unlit (blue) conditions.**
An asterisk denotes statistically clear differences between the two treatments, whereas N.S. denotes results
lacking statistical clarity. (A) The number of spiders per bridge panel. (B) The web catch area (cm2) of
spiders in both light conditions. (C) Body condition within the two light conditions (as measured by the
residual index: residuals of linear model pitting weight against femur length as (outlined in *Jakob, Mar-
shall & Uetz, 1996*). Body condition is an accepted proxy for reproductive fitness in spiders, thus indicat-
ing that spiders fare better in artificially lit conditions. (D) Body size between treatments. There is no sta-
tistically clear difference in body size between the treatments.

the number of prey captured are expected to increase with an increase in web catch area,
thus, web area was also used as a covariate in prey models as well.

## RESULTS

### Spider web abundance

There were significantly more spiders present in unlit (mean = 14, SD = 5.94, $n = 44$,
range = 3–28) panels than in lit panels (mean = 10.1, SD = 4.89, $n = 44$, range = 4 - 24;
GLM: $z\ value = 3.35$, $df_{residual} = 85$, $p = 0.0008$: parameter estimate from unlit to lit is
1.38, 95% CI [1.14–1.66]; Fig. 3A).

### Web structure

Orb web dimensions differed between lit and unlit conditions. Webs in unlit panels had
a mean catch area of 639 cm$^2$ (SD = 317, $n = 43$), while those in lit conditions (mean =
414 cm$^2$, SD = 285, $n = 43$) were smaller (GLM: $t\ value = 3.35$, $df_{residual} = 84$, $p = 0.001$;
model parameter estimate from unlit to lit = 1.54, 95% CI [1.20–1.99]; Fig. 3B).

The effect of light on vertical web asymmetry, was not statistically significant (LM:
$F_{2,83} = 3.433$; effect of light: $t\ value = 1.54$, $p = 0.13$; model parameter estimate from unlit
to lit = 0.06, 95% CI [−0.02 –0.14]; Fig. 4; webs in lit conditions: mean = −0.13, SD =

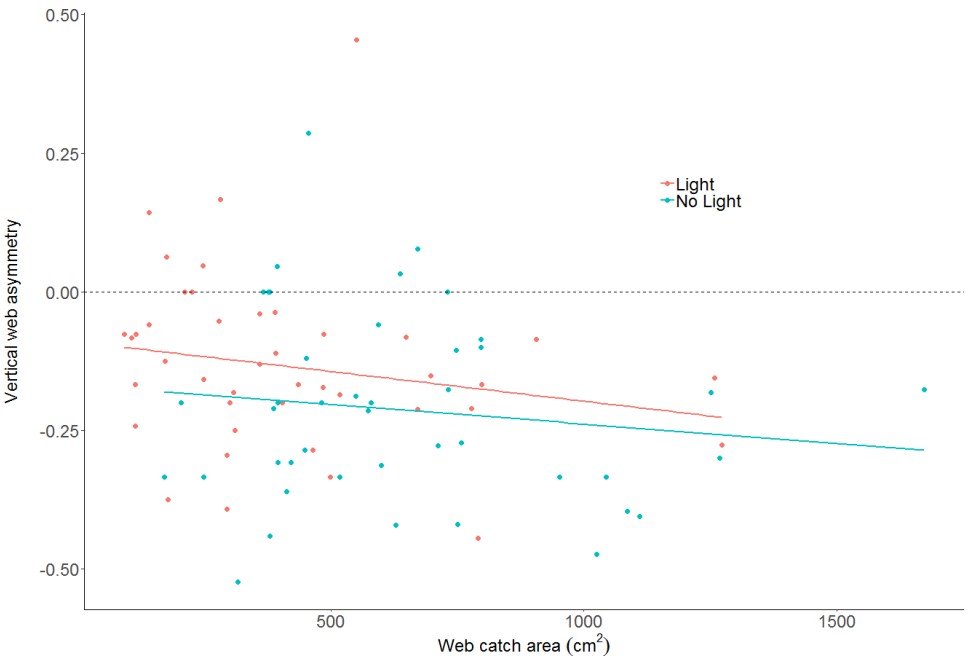

**Figure 4** **Plot of vertical web asymmetry (VWA) as a function of web area, in the two light conditions.**
The dotted line shows a perfectly symmetrical web at 0. A linear model is plotted through the data as
trend-lines. Neither light conditions, nor area, are statistically clear predictors of VWA.

0.16, $n = 43$; webs in unlit conditions: mean = $-0.21$, SD = 0.17, $n = 43$). Additionally,
vertical web asymmetry did not appear to differ with changes in web catch area within the
same model (effect of web area: *t value* = $-1.44$, $p = 0.15$), as was previously found for
some spiders of the Family Tetragnathidae (*Tew & Hesselberg, 2018*).

### Prey capture and body condition

The number of prey items caught in webs increased in the light condition (GLM: *z value*
= 10.23, *df* $_{residual}$ = 49, $p < 0.0001$; parameter estimate from unlit to lit conditions = 6.94,
95% CI [4.79–10.06]; Fig. 5), while also increasing with larger web area (GLM: *z value* =
4.19, *df* $_{residual}$ = 49, $p < 0.0001$). Webs in lit conditions had, on average, over 30 more prey
(mean = 39.2, SD = 25.6, $n = 26$), than webs in unlit conditions (mean = 7.78, SD = 7.3,
$n = 27$).

Body condition was significantly higher in spiders that were found in lit conditions
(mean = 0.02, SD = 0.04, $n = 23$) than spiders found in unlit conditions (mean = $^-0.04$,
SD = 0.03, $n = 15$; LM: $F_{1,36} = 22.84$; effect of light: *t value* = $-4.78$, $p < 0.0001$; model
parameter estimate from unlit to lit = $-0.06$, 95% CI [$-0.03$ to $-0.08$]; Fig. 3C). Body
size (measured by femur length), on the other hand, was not different for spiders found in
lit conditions (mean = 5.03 mm, SD = 0.49, $n = 23$) compared to spiders found in unlit
conditions (mean = 5.00 mm, SD = 0.41, $n = 15$; LM: $F_{1,36} = 0.03$; effect of light: *t* value
= $-0.17$, $p = 0.87$; Fig. 3D).
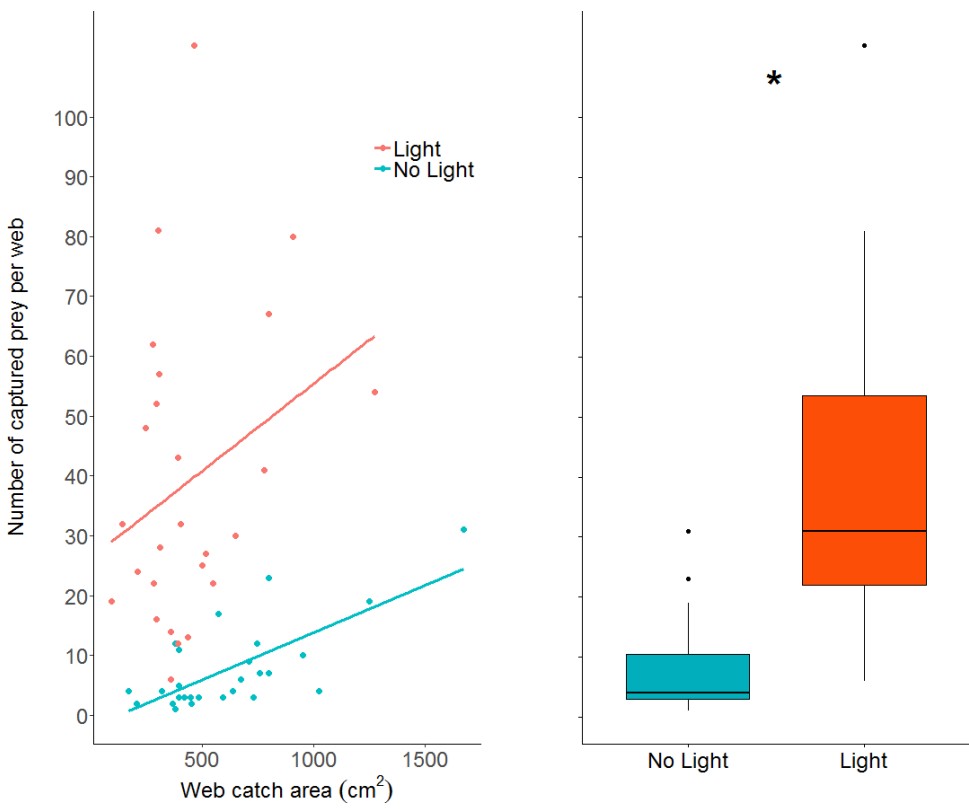

**Figure 5  A dot plot of the number of prey captured per web, grouped by light condition and plotted against web catch area.** There is a statistically clear effect of web catch area and light treatment on prey capture, denoted by the asterisk. A simple linear model was plotted through the data as trend-lines for visualization. Boxplot shows the same data without the regression across the x axis (web catch area is collapsed).

## DISCUSSION

In this case study, anthropogenic night lighting had significant, important effects on the abundance, web structure, capture success, and body condition of orb-weaving spiders on one bridge in France. Fewer spiders *(Larinioides sclopetarius)* were found in lit bridge panels than those in unlit conditions, and those that built webs under night lighting had significantly smaller webs. However, webs in lit conditions caught significantly more prey than larger webs in unlit areas. Additionally, spiders in lit conditions had significantly higher body conditions than those in unlit conditions.

*L. sclopetarius* that built webs in lit conditions were able to capture more insects using significantly smaller web structures, a result consistent with previous findings of *Neoscona crucifera (Araneidae)* in natural moonlight conditions (*Adams, 2000*), yet conflicts with recent studies of *Nephila pilipes (Araneidae)* prey interception in artificial light (*Yuen & Bonebrake, 2017*). This suggests that they have an energetic advantage over individuals in unlit conditions, as other studies have shown that building smaller orb webs and catching more prey leads to increases in growth and fecundity (*Wise, 1975*; *Higgins, 1995*;

*Jakob, Marshall & Uetz, 1996*; *Kreiter & Wise, 2001*). Similarly, previous research with *L. sclopetarius* has shown that growth of these spiders is highly dependent on food availability (*Kleinteich & Schneider, 2011*). This may explain my finding that spiders in lit areas had significantly higher body condition than those in unlit areas (since they had captured more prey). This can have important consequences for lifetime reproductive success, as food limited *L. sclopetarius* have lower clutch sizes (*Kleinteich, Wilder & Schneider, 2015*). However, it is important to note that prey availability within the two conditions were not measured. Thus, I cannot exclude the possibility that webs in lit conditions are constructed differently (e.g., more sticky silk, higher tensile strength, etc.), and are better suited at catching equally-available prey. Furthermore, it is possible that spiders caught more prey in lit areas because there were fewer spiders with webs to compete with. Building a web in a location with less competition could lead to higher prey capture success even if prey availability is similar.

Despite the apparent energetic benefits derived from building webs in lit areas, I found that spider abundance was lower in lit conditions, with more spiders found in unlit panels —similar to the results found by *Meyer & Sullivan (2013)* with Tetragnathid spiders in LED light. There are a few potential explanations for this result. One possibility may be that most of the spiders in lit panels were satiated, and did not have an active orb web. Spiders that weren't 'hungry' and waiting on a web's hub or radii were not counted. Thus, it may appear that there are fewer spiders in lit areas, when there are actually fewer actively foraging spiders.

Previous research on a closely-related species (*Larinioides cornutus*) showed that satiated spiders invest less in web building (*Sherman, 1994*), likely because it is energetically expensive (*Opell, 1998*). However, other orb-weavers (*Zygiella x-notata;* also Family Araneidae) build webs more frequently when they have a nutrient-rich diet available, presumably because they have more energy reserves to devote to web building (*Mayntz, Toft & Vollrath, 2009*). In this latter case, prey quality appears to be more important than prey quantity. Thus, a second possibility is that *L. sclopetarius,* in the present study, is less common in lit areas because prey quality has decreased in these same areas due to artificial lighting.

Thirdly, it is possible that spiders in these profitable areas are dominant, and are successful at keeping others out of their territories. *Heiling & Herberstein (1999b)* showed that larger individuals of *L. sclopetarius* are able to keep smaller individuals out of 'prime' territories, although they found an overall increase in spider density in lit areas, rather than the decrease that I found. This may be due to the short duration of the current study, as opposed to the much longer duration study by *Heiling & Herberstein (1999b)*. Alternatively, this discrepancy may simply be due to survey methods. *Heiling & Herberstein (1999b)* counted all female *L. sclopetarius*, whereas I only counted adult female *L. sclopetarius* that were actively waiting on or near an orb-web.

Fourthly, spiders may be avoiding this seemingly advantageous area because of their own risk of becoming prey. While spiders likely benefit in lit conditions from increased prey capture, predators of spiders may have similar advantages. Birds, for example, are known to increase nocturnal activity, increase foraging effort, increase use of vision for

foraging, and have improved prey capture rates in artificially lit areas (*Santos et al., 2010*; *Dwyer et al., 2013*; *Russ, Rüger & Klenke, 2015*). This may pose an actual predation risk or simply create local peaks of perceived predation risk within a landscape of fear for spiders (*Schmitz, Beckerman & O'Brien, 1997*; *Schmitz, Krivan & Ovadia, 2004*).

Lastly, it is possible that artificial light has caused unmeasured negative physiological effects on these spiders that increase mortality directly or indirectly. For example, ALAN can impair immune function (*Durrant et al., 2015*) and inhibit diapause (*Shah et al., 2011*), either of which could have negative survival consequences. Clearly, more mechanistic and population-level studies are required to understand the effects of artificial lights on spiders, predator–prey relationships (with spiders as both predators and prey), and food webs in general, especially within an evolutionary framework (*Swaddle et al., 2015*; *Czaczkes et al., 2018*). These common bridge spiders are closely associated with humans and urban landscapes, thus an interesting next step would be to understand how these spiders might be evolving in response to novel sensory environments (such as artificial light).

## CONCLUSION

In a natural artificial light experiment, *Larinioides sclopetarius* were less active, and had smaller webs in artificially lit areas. Yet, it appears that *L. sclopetarius* benefit by catching more prey and having higher body condition. However, it is important to note that this natural experiment occurred in one location at one 'snapshot' in time. It is possible that similar data collection across many bridges, and across longer timescales will yield different results than found here. Additionally, whether or not spider fitness or populations are affected is still unclear. Predators, such as spiders, may be able to exploit particular prey in artificial light conditions, but is this urban trap sustainable?

## ACKNOWLEDGEMENTS

I would like to thank Robert Wells and Marie-Christine Wells for logistical support, Jesse R. Barber, Thomas Hesselberg, Cory A. Toth, Cristina Barber, Christine D. Hayes, Juliette J. Rubin, Timothy Bonebrake, and an anonymous reviewer for valuable comments on earlier versions of the manuscript. Raw data and R code are available on Zenodo (10.5281/zenodo.3698188). Voucher specimens of *L. sclopetarius* are deposited in the Entomology Collections at the California Academy of Sciences.

### Funding

Dylan G.E. Gomes is funded by the National Science Foundation's Graduate Research Fellowship Program, while research was supported by Boise State University and their Ecology, Evolution, and Behavior Program. The funders had no role in study design, data collection and analysis, decision to publish, or preparation of the manuscript.

## Grant Disclosures

The following grant information was disclosed by the author:
National Science Foundation's Graduate Research Fellowship Program.
Boise State University and their Ecology, Evolution, and Behavior Program.

## Competing Interests

The author declares there are no competing interests.

## Author Contributions

- Dylan G.E. Gomes conceived and designed the experiments, performed the experiments, analyzed the data, prepared figures and/or tables, authored or reviewed drafts of the paper, and approved the final draft.

## Data Availability

All data and code are available on Zenodo (10.5281/zenodo.3698188).

All voucher specimens are stored at the California Academy of Sciences in their Entomological Collections: CASENT8413329 and CASENT8413330

## Supplemental Information

Supplemental information for this article can be found online at http://dx.doi.org/10.7717/peerj.8808#supplemental-information.

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
