# Peer review of "Orb-weaving spiders are fewer but larger and catch more prey in lit bridge panels from a natural artificial light experiment"

_PeerJ, doi:10.7717/peerj.8808_

## Round 0.1 · original submission · Major Revisions

We have received the reviews from two reviewers, and both of them have suggested major revisions. Please make the revisions and resubmit.

·

Basic reporting

The article is well written and presented.

Experimental design

As detailed more fully in my general comments, I have some questions as to the independence of the panels/sites and thus the general validity of the statistical results.

Validity of the findings

As also detailed in my general comments, there is some over-speculation or over-generalizations, particularly with respect to the title and abstract.

Additional comments

The author exploits a natural experiment in the form of a bridge with alternating lit and unlit panels to explore how the lighting affects spiders living on those panels. The results appear to show more spiders in unlit panels and larger spiders in lit panels. The strongest result may be the larger number of captured prey in lit webs vs unlit webs, particularly after controlling for web size. I think these results are likely robust and could contribute a nice piece to the growing work on light and urban spiders. I do have a few major concerns however:

First, there is a real statistical issue with respect to independence. Each panel is treated as an “independent group” but this seems difficult to justify. They’re all right next to one another, right? I would guess that spatial autocorrelation could be an issue here and that other spatial issues might be apparent. Are there patterns with respect to whether a spider or panel is in the middle of the bridge vs at the end? I recommend further explanation and examination of these aspects in the methods. But most critically, there is one bridge and so one site really. In the conclusion it is noted that the study represents “one location” and one “snapshot”. So, I appreciate that this is recognized but I think that this should be better articulated; how might these results change if the study went on for longer than a week or if multiple bridges were used? Expansion of this in the discussion would be helpful.

Second, additional natural history details are needed. It’s not entirely obvious but it seems like all of the studied spiders were Larinioides sclopetarius. Can you provide more details about the species? There’s actually quite of bit work on this system within which it would be helpful to contextualize your results:
Kleinteich, A., Wilder, S. M., & Schneider, J. M. (2015) Contributions of juvenile and adult diet to the lifetime reproductive success and lifespan of a spider. Oikos, 124, 130-138. [this one is particularly relevant given the implications for food restriction]
Kleinteich, A., & Schneider, J. M. (2011) Developmental strategies in an invasive spider: constraints and plasticity. Ecological Entomology, 36, 82-93.
And also, the Heiling studies already cited in the paper. At present it’s hard to link the results to what this means for the species or ecology. Greater engagement in this regard would be helpful.

Third, particularly given the first point above, I urge greater caution and constraint in how the results are presented and interpreted. For example, the title and abstract are much too broad given the study (more detail on this below).

Additional comments:
Title – needs to be better constrained and described… at present it reads like a review which this is not, best to have a title that captures the essential result and in the proper context, e.g. “Orb-weaver spiders are fewer and smaller in unlit bridge panels from a natural artificial light experiment” (or something like this)
Abstract – at present, about half of the abstract is speculative… I would reduce this in favor of focusing more on the methods and results (there is essentially nothing in the abstract about the methods)
line 10 – the relationship between insects and light is complex, e.g.
Owens, A. C., & Lewis, S. M. (2018). The impact of artificial light at night on nocturnal insects: A review and synthesis. Ecology and Evolution, 8, 11337-11358.
I think this paragraph and the following two could be better developed
line 36 – is a week of data enough?
line 40 – what’s the light brightness? Is it fluorescent? More details needed here regarding the light source
line 42 – “unlit”
line 55 – so were all the species recorded (not just collected) Larinioides sclopetarius? And if so, why not make this clear in the abstract? If not, what else was observed?
Line 75 – what was the prey? And is number of prey most appropriate here instead of say, prey mass?
lines 106-108 – what’s the range of spiders in lit and unlit?
line 114 – I would interpret this as there not being effect (it isn’t a problem of the statistics)
line 120 – what family?
line 152 – I presume that the idea here is that lit webs are attracting more prey. This is discussed in the intro but should be made explicit here as well. But also, there are more webs in unlit conditions, thus it could just be that there are more webs competing for the same number of insect prey and webs in lit conditions take advantage of this (rather than there simply being more prey)
line 169 – I don’t think it’s very clear, particularly given that you didn’t measure prey. The main point of this sentence is fine but I would suggest being cautious about claims like this.

Figs 3-5: Increase font size of axes to improve readability

Reviewer 2 ·

Basic reporting

This is a well-written manuscript that clearly explains the experiment using appropriate language. The figures are relevant and easy to understand. The literature relating to spiders foraging under artificial light at night is well referenced. However, the introduction requires a lot more context for the effects of light pollution broadly, particularly for its physiological effects which have potential consequences for body condition and mortality, both of which are directly relevant to this experiment. More information is also needed for the basic biology of web building, and the importance of different lighting technologies to provide context to these results.

Experimental design

This is a well-designed and simple experiment that neatly tests a hypothesis, and these connections are clearly communicated. The introduction provides context for why investigating web building, abundance and body condition of an orb-weaver in the field is important. Some of the methods require more description, such as the choice of web for architecture measurements, and better justification is required for the use of body condition residuals. It would also be useful to explain why this paper is important given that a similar experiment was performed by Heiling & Herberstein (1999).

Validity of the findings

The discussion provides a useful interpretation of the results, supported by the literature. This would be improved by expanding on explanations of results, given improved context in the introduction (e.g. physiological damage from ALAN), and providing better comparisons with cited literature (e.g. differences with Zygiella, similar result found for tetragnathids, effects of light type, etc.).

Additional comments

This is a very interesting study, and you have done a good job communicating your results and linking your results to your hypotheses. My main suggestions for improvement are to provide a more comprehensive introduction and a more cohesive discussion.

Annotated reviews are not available for download in order to protect the identity of reviewers who chose to remain anonymous.

---

## Round 0.2 · Minor Revisions

Please make the remaining minor revisions suggested by Reviewer 1.

·

Basic reporting

The revised version is clear and now well placed in the context of the literature and with respect to the implications of the results.

Experimental design

The additional detail provided in the revision is key and I'm satisfied with the justification and scope of the results.

Validity of the findings

Results are clear and well explained.

Additional comments

The revised version is much improved and the changes are well explained. In the revision the results are much clearer and the possible causes behind them more reasonable and complete. At this point I don’t have any further issues with the work and feel that it should serve as a useful contribution to the literature. I have a few suggestions and edits below – but there IS the important matter of Fig. 3 within which I think an unintentional error was introduced during revision.

Abstract – “not as active within these habitats” – this isn’t a landscape level effect is it?
Line 75 – any need for identifying body condition as a proxy for fitness? If not, I would delete this.
Line 92 – missed return for the subheading
Line 131 – microscope?
Line 183 – can remove “in them”
Fig 3 – something went wrong here, some of the asterisks were replaced by N.S.s

Reviewer 2 ·

Basic reporting

I am happy with how the manuscript has been restructured

Experimental design

The experimental design is now easier to follow, and I am satisfied with the author's responses to my queries

Validity of the findings

The data have been better placed in the appropriate context, and the conclusions are now better supported by the rest of the manuscript

---

## Round 0.3 · accepted · Accept

Thanks for all your hard work on the paper. It is now accepted for publication.